# Irreversible fate commitment in the *Arabidopsis* stomatal lineage requires a FAMA and RETINOBLASTOMA-RELATED module

**Juliana L Matos[1†], On Sun Lau[1†], Charles Hachez[1†‡], Alfredo Cruz-Ramírez[2§], Ben Scheres[2¶], Dominique C Bergmann[1,3]***

[1]Department of Biology, Stanford University, Stanford, United States; [2]Department of Molecular Genetics, Utrecht University, Utrecht, Netherlands; [3]Howard Hughes Medical Institute, Stanford University, Stanford, United States

**\*For correspondence:** bergmann@stanford.edu

[†]These authors contributed equally to this work

**Present address:** [‡]Institut des Sciences de la Vie, Université Catholique de Louvain, Louvain-la-Neuve, Belgium; [§]Laboratorio Nacional de Genómica para la Biodiversidad, CINVESTAV, Irapuato, Mexico; [¶]Department of Plant Developmental Biology, Wageningen University, Wageningen, Netherlands

**Reviewing editor**: Richard Amasino, University of Wisconsin, United States

**Abstract** The presumed totipotency of plant cells leads to questions about how specific stem cell lineages and terminal fates could be established. In the *Arabidopsis* stomatal lineage, a transient self-renewing phase creates precursors that differentiate into one of two epidermal cell types, guard cells or pavement cells. We found that irreversible differentiation of guard cells involves RETINOBLASTOMA-RELATED (RBR) recruitment to regulatory regions of master regulators of stomatal initiation, facilitated through interaction with a terminal stomatal lineage transcription factor, FAMA. Disrupting physical interactions between FAMA and RBR preferentially reveals the role of RBR in enforcing fate commitment over its role in cell-cycle control in this developmental context. Analysis of the phenotypes linked to the modulation of FAMA and RBR sheds new light on the way iterative divisions and terminal differentiation are coordinately regulated in a plant stem-cell lineage.

## Introduction

Plants exhibit remarkable developmental plasticity and their cells are typically considered totipotent, in that a complete plant can be regenerated from nearly any isolated individual cell. In intact plants, however, distinct cell lineages emerge and terminal fates are stable. A prime example of a specialized lineage is in the *Arabidopsis* leaf epidermis (*Figure 1A*) where asymmetric divisions of protodermal cells generate meristemoid mother cells (MMC) and meristemoids (M), self-renewing cells akin to transit amplifying cells in mammalian stem cell lineages (*Lau and Bergmann, 2012*; *Pillitteri and Dong, 2013*). At the end of their renewing stages, these meristemoids differentiate into guard mother cells (GMCs), which undergo a single symmetric division to generate the paired guard cells (GCs) of the mature stomata. GCs and each of the intermediate stages leading to their formation are characterized by distinct morphologies and unique gene expression profiles, allowing experimental dissection of lineage progression in intact, developing organs (*Lau and Bergmann, 2012*; *Pillitteri and Dong, 2013*).

The basic helix-loop-helix (bHLH) transcription factor FAMA is a master regulator of guard cell identity; it is necessary and sufficient for GC fate acquisition and its epidermal expression is limited to GMCs and young GCs (*Ohashi-Ito and Bergmann, 2006*) and (*Figure 1B*). GMCs are made in *fama* mutants, but they fail to progress into GCs and instead continue dividing while maintaining expression of earlier fate markers (*Ohashi-Ito and Bergmann, 2006*) and (*Figure 1B*, inset); this failure to make GCs results in seedling lethality (*Ohashi-Ito and Bergmann, 2006*) and (*Figure 1I*). Overexpression of FAMA reprograms other cells into GC identity, while simultaneously repressing cell division to yield

**eLife digest** Stem cells in animals and plants help to make and replenish the tissues of the body by dividing and becoming specialized types of cells. Once specialized for a certain function, it is important that a cell keeps that function. In plant leaves, one type of stem cell makes two different types of specialized cells: pavement cells and stomatal guard cells. Pavement cells lock together to form a waterproof barrier to the outside, while guard cells surround the small pores that open and close to allow the plant to exchange water, oxygen and carbon dioxide with the atmosphere.

Once a cell becomes a pavement cell or a guard cell, it does not change its identity again. However, if a single cell is removed from a plant, it can revert to a stem cell and a whole new plant can be grown from it. This poses the question of how, in intact plants, specialized cells like pavement cells and guard cells are prevented from reverting to stem cells.

In *Arabidopsis thaliana*, a small flowering plant that is widely used as a model organism in research, a protein called FAMA is responsible for controlling a set of genes that turn stem cells into guard cells. Matos et al. have now found that FAMA needs to bind to another protein called RBR to control this process. It seems that these two proteins make the transition from stem cell to guard cell permanent by changing the structure of DNA in regions that control stem cell genes.

RBR is similar to a human protein called Retinoblastoma that helps prevent tumors and regulate stem cells, but how it actually performs these functions in humans is still debated. Because stem cells and guard cells are displayed on the surface of plant leaves and leave behind clues of their past, Matos et al. were able to watch stem cells grow up to be mature guard cells. When the partnership between FAMA and RBR was broken, it was possible to watch those same guard cells revert backwards into stem cells. Seeing development 'rewind' could provide useful insights into the way in which cell identity is controlled in both plants and animals.

single-celled stomata (*Ohashi-Ito and Bergmann, 2006*). The mechanisms by which FAMA regulates cell division and terminal differentiation are not known, but FAMA's direct targets include cell cycle regulators and genes associated with mature guard cell function (*Hachez et al., 2011*). FAMA has been shown to act as a transcriptional activator (*Ohashi-Ito and Bergmann, 2006*) but can also participate in repression of certain cell cycle targets (*Hachez et al., 2011*). Here we show that FAMA is required for the irreversible differentiation of GCs and that it fulfills this role through recruitment of the *Arabidopsis* Retinoblastoma homologue, RETINOBLASTOMA-RELATED (RBR). Point mutations that disrupt FAMA-RBR interactions render FAMA capable of promoting initial GC identity, but unable to maintain commitment. By demonstrating FAMA-promoted binding of RBR to the regulatory regions of stomatal regulators whose genomic regions contain repressive chromatin marks, we define a molecular mechanism by which the ubiquitously expressed RBR is recruited to specific genomic contexts at specific times to regulate key developmental events.

## Results

RBR is broadly expressed in *Arabidopsis* development and reduction of RBR activity has been correlated with excess division and loss of cell identity in many different contexts, including the early stomatal lineage (*Borghi et al., 2010*). In the epidermis of actively dividing young leaves, RBRp:RBR-CFP (*Cruz-Ramirez et al., 2012*) is expressed in all cell nuclei; as the leaf matures, expression becomes restricted to stomatal lineage cells (*Figure 1C*). Mosaic co-suppression of the *RBRp:RBR-CFP* transgene leads to loss of fluorescence and concomitant excessive divisions in the CFP-minus sectors, suggesting that RBR represses cell divisions in both the early lineage and the terminally differentiated GCs (*Figure 1D*). To examine RBR's role specifically in the GCs, we drove expression of artificial microRNAs (amiRNAs) against RBR by the FAMA promoter. *FAMAp:amiRNA-RBR* GCs underwent inappropriate extra divisions oriented transverse to the long axis of the cells, while other epidermal cells were not affected, confirming a direct requirement for RBR in GCs (*Figure 1E* and *Figure 1—figure supplement 1A*) and confirming phenotypes reported using different amiRNAs directed against RBR (*Lee et al., 2014a*).

FAMA encodes a canonical RBR binding motif (LxCxE) (*Burkhart and Sage, 2008*) that is conserved among dicot FAMA orthologs, but not in FAMA's closest paralogs SPEECHLESS (SPCH) and MUTE

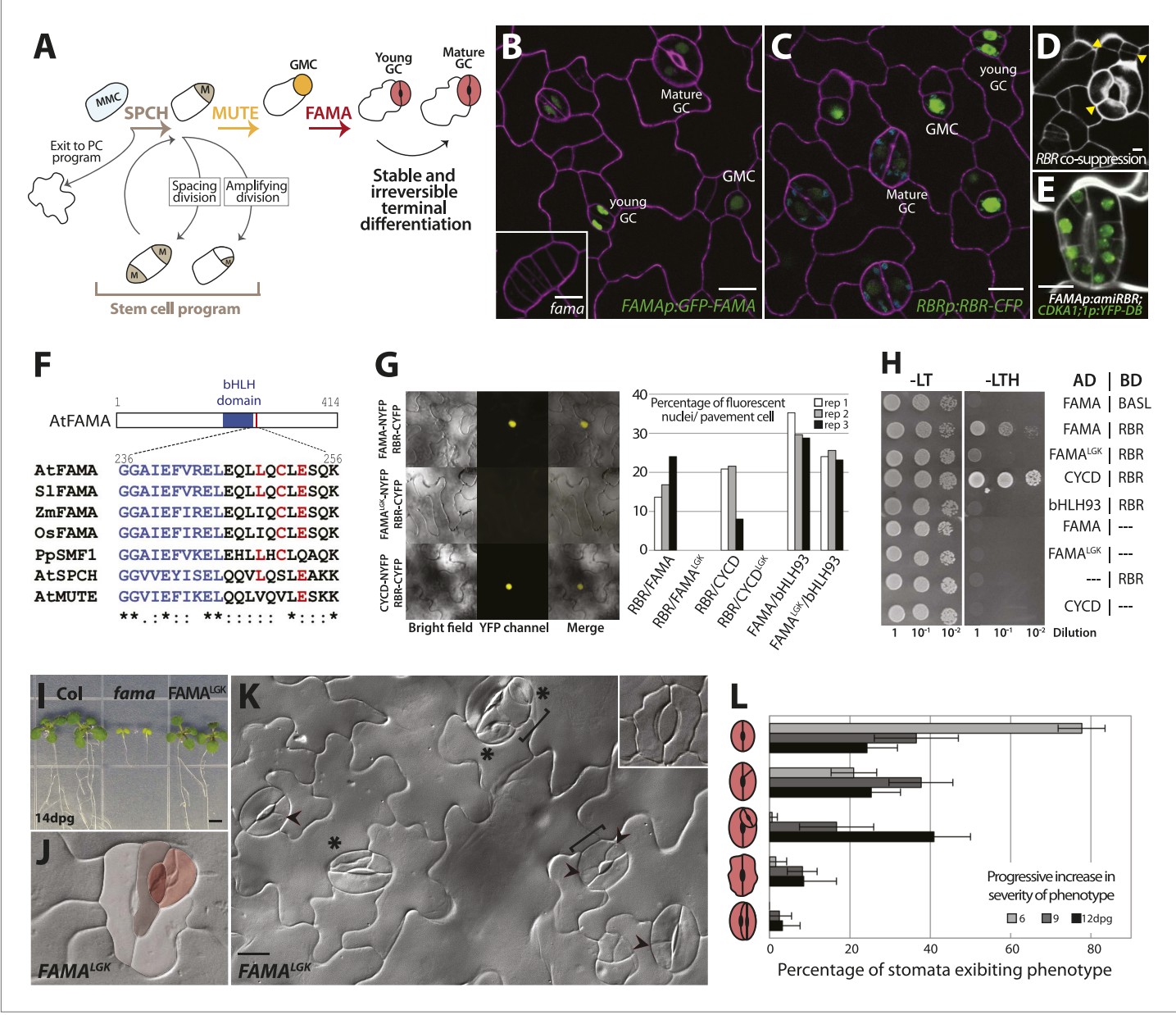

**Figure 1**. FAMA and RBR physically interact and regulate guard cell division and differentiation. (**A**) Schematic of key stages in stomatal development mediated by bHLHs SPCH, MUTE and FAMA. Cell types are labeled as: meristemoid mother cell (MMC), meristemoid (M), stomatal lineage ground cell (SLGC), guard mother cell (GMC), guard cell (GC), pavement cell (PC). (**B** and **C**) Expression of FAMA and RBR in GMCs and GCs. Confocal images of 5-days post germination (dpg) cotyledon of FAMAp:GFP-FAMA (**B**, in green) and RBRp:RBR-CFP (**C**, in green). Inset in (**B**) is a *fama* mutant GMC at 10-dpg. Cell outlines (purple) were visualized with propidium iodide. (**D** and **E**) Reduction in RBR level leads to extra divisions in GCs. Confocal images of a co-suppressed RBRp:RBR-CFP line (**D**) and *FAMAp:amiRBR* expressing a CDKA1;1 reporter (green) (**E**). Yellow arrowheads in (**D**) indicate ectopic cell divisions. Cell outlines (white) were visualized with propidium iodide. (**F**) ClustalW2-based protein alignment of the LxCxE motif among FAMA relatives. (**G** and **H**) FAMA interacts with RBR in vivo and in vitro through its LxCxE motif. Representative images (**G**, left) and quantified data (**G**, right; rep: replicate) of Bimolecular Fluorescence Complementation (BIFC) analysis between FAMA and RBR. Pairs of CYCD/CYCD^LGK-RBR and FAMA-bHLH93 (***Ohashi-Ito and Bergmann, 2006***) were used as controls. (**H**) Yeast two-hybrid interaction assays between FAMA and RBR. (**I**) Complementation of seedling lethality in *fama* mutants by FAMA^LGK (*FAMAp:FAMA^LGK*; *fama*). (**J**–**L**) Diversity of GC defects in adaxial cotyledon epidermis of 12-dpg FAMA^LGK. DIC images of a mature GC showing strong phenotype (**J**, false red colors indicate different GC units within another) and a broader view of GCs with different defects (**K**). Key: ectopic asymmetric divisions (arrowheads), new GC units (asterisks), properly spaced divisions and GC units (brackets). Inset shows a lobed GC. (**L**) Quantitation of different classes of GC defects (cartoons on Y-axis) in FAMA^LGK at 6, 9 and 12-dpg. Bars represent the percentages of each class over all GCs on adaxial cotyledons. All images are at the same magnification (including insets in **B** and **K**). Scale bar, 10 μm.

*Figure 1. Continued on next page*

*Figure 1. Continued*

The following figure supplements are available for figure 1:

**Figure supplement 1**. Additional images of FAMA promoter-driven expression of amiRBR and of FAMA<sup>LGK</sup>-YFP.

**Figure supplement 2**. Categorization of guard cell (GC) defects and increase in severity over time in FAMA<sup>LGK</sup>.

(*Figure 1F*). LxCxE-dependent physical interaction between FAMA and RBR was tested by *in planta* Bimolecular Fluorescence Complementation (BiFC) (*Figure 1G*) and yeast two-hybrid (*Figure 1H*) assays. In both assays, WT FAMA, but not a version bearing point mutations changing the Cysteine (C) and Glutamate (E) in the LxCxE motif to Glycine (G) and Lysine (K) (FAMA$^{LGK}$) could interact with RBR. Importantly, FAMA$^{LGK}$ could still interact with its dimerization partner bHLH93 (*Ohashi-Ito and Bergmann, 2006*) (*Figure 1G*), indicating that the FAMA$^{LGK}$ variant maintains its overall structural integrity.

We then asked whether physical interaction with RBR was required for FAMA function in the context of normal leaf development. *FAMAp:FAMA$^{LGK}$* was tested for its ability to complement *fama* lethality and defects in GC differentiation, and *FAMAp:FAMA$^{LGK}$-YFP* was monitored to determine whether the LCE→LGK modification altered FAMA's expression, stability or subcellular localization. In young cotyledons and leaves, *FAMAp:FAMA$^{LGK}$-YFP* was exclusively nuclear. Like G/YFP-tagged versions of FAMA published previously (*Ohashi-Ito and Bergmann, 2006*; *Pillitteri et al., 2007*; *Lee et al., 2014a*), *FAMAp:FAMA$^{LGK}$-YFP* is first apparent in GMCs, remains highly expressed as the GMCs undergo cell division, and is downregulated as GCs mature such that stomata with clearly defined pores express the protein at low levels or not at all (*Figure 1B* and *Figure 1—figure supplement 1B–C*). Plants of genotype *fama;FAMApro:FAMA$^{LGK}$* (hereafter referred to as FAMA$^{LGK}$ plants) were recovered and were moderately healthy and fertile, though smaller than wild type, indicating that FAMA$^{LGK}$ was sufficient to rescue *fama* lethality (*Figure 1I*). In the GCs of rescued FAMA$^{LGK}$ plants, however, we observed excessive cell divisions, changes in cell morphology, and, most strikingly, production of paired GCs inside of existing GCs (*Figure 1J–L* and *Figure 1—figure supplement 2*, phenotypic classes 6–11).

Phenotypes conferred by FAMA$^{LGK}$ and by manipulating RBR in the late stomatal lineage both involved increased cell division, but were not identical. To improve phenotypic resolution, we characterized the expression patterns of cell fate and cell cycle markers in FAMA$^{LGK}$ and *FAMAp:amiRBR* plants (*Figure 2* and *Figure 2—figure supplement 1*). This detailed analysis revealed clear phenotypic differences between reducing RBR levels in GMCs and reducing RBR's interaction with FAMA (*Figure 2—figure supplement 1A–B*). Notably, the FAMA$^{LGK}$ phenotype results, not from chaotic or uncontrolled divisions and fate changes, but rather an orderly reiteration of stomatal lineage progression. This manifested itself as a progressive increase in phenotypic severity with age (*Figure 1L*) and by the appearance of stomatal lineage markers in patterns suggesting that the GCs reverted to MMC identity and proceeded through the intermediate stages of the pathway normally (*Figure 2*). Expression of stomatal-promoting transcription factors (SPCH, MUTE, FAMA, *Figure 2A–C* and *Figure 2—figure supplement 2*), stomatal-restricting signaling elements (TMM, EPF1, EPF2, *Figure 3A–D*), and general division reporters (*CDKA1;1*, *Figure 2B–C*) followed the normal temporal patterns, and ectopic GC divisions appeared to follow early lineage division rules. For example, when a 'reprogrammed' GC produced two stomata, they were separated by a non-stomatal cell, indicating that spacing divisions occurred. Distinct cell orientations characteristic of amplifying divisions were also visible (*Figure 2B* and *Figure 2—figure supplement 3*). Further evidence for normal asymmetric divisions is polarized localization of BASL (*Dong et al., 2009*) in the larger daughter of a GC division (*Figure 3E*). Based on the lack of expression of stomatal lineage markers (*Figure 2D*), we interpret the lobed GCs we observe at low, but significant, frequencies in FAMA$^{LGK}$ plants (*Figure 1K*, inset, and *Figure 1L*) as cells that are transdifferentiating into an epidermal pavement cell identity.

In *FAMAp:amiRBR* lines, by contrast, excessive GC division was accompanied by elevation of cell cycle gene expression throughout the GCs (CDKA1;1, *Figure 1E* , CDKB1;1 and CDC6, *Figure 2—figure supplement 1A*), but only rarely by misexpression of early stomatal lineage markers (*Figure 2—figure supplement 2B*). Consistent with gene expression behaviors, spacing and amplifying divisions were not seen in *FAMAp:amiRBR* cotyledon GCs at any appreciable frequency (<1/1000 GCs) in 6–12 day

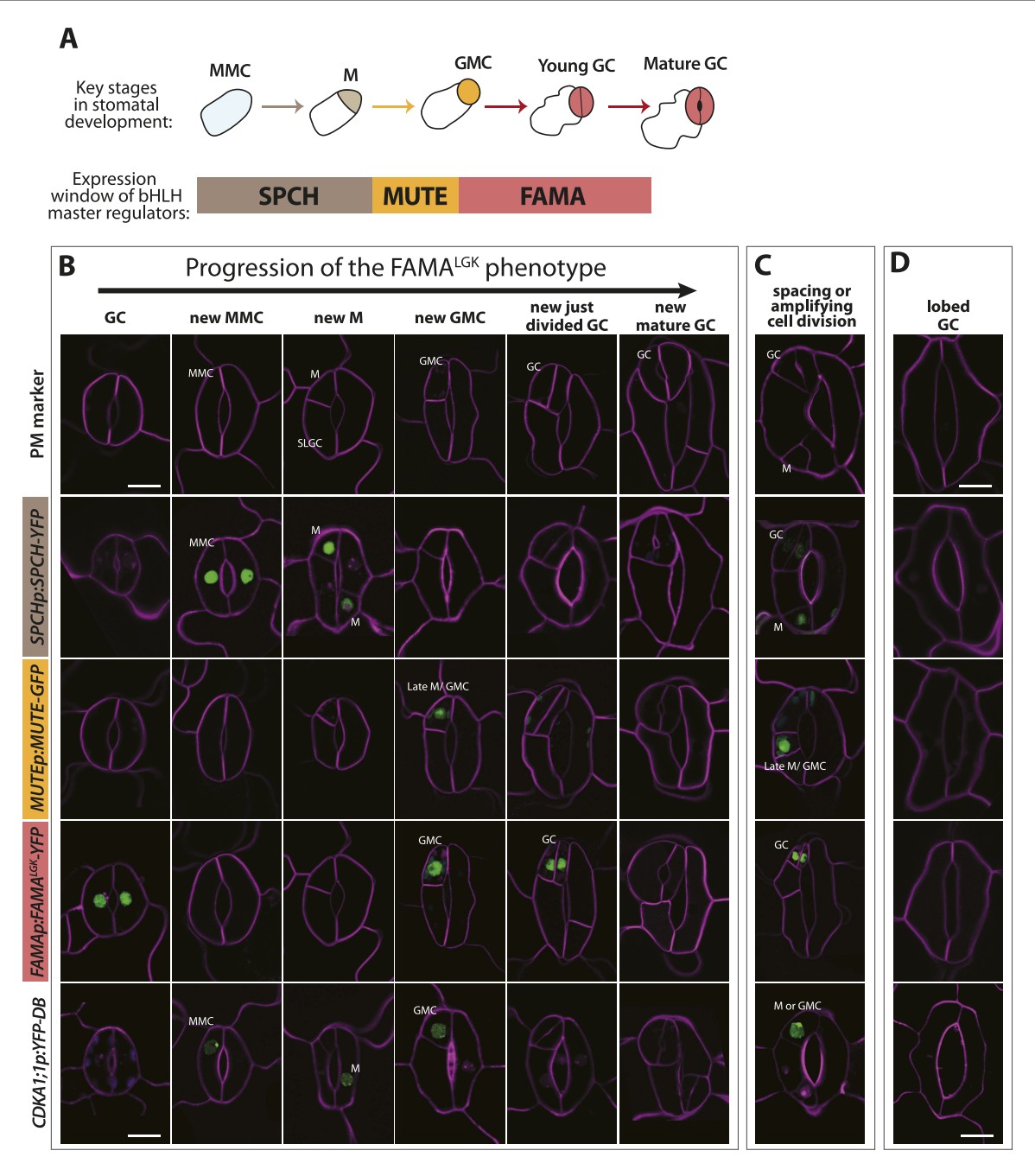

**Figure 2**. Disruption of FAMA-RBR interaction leads to failure of terminal differentiation and reiteration of stomatal lineage divisions and gene expression programs. (**A**) Diagram of stages of stomatal development (abbreviated and color-coded as in *Figure 1A*) and expression window of bHLH transcription factors SPCH, MUTE and FAMA. (**B**) Characterization of GC defects in FAMA[LGK] plants accompanied by key stomatal reporters. Wild type-looking GCs of FAMA[LGK] plants re-iterate the stomatal developmental pathway, undergo further divisions and exhibit correct orderly expression of stage-specific stomatal regulators and cell cycle genes. Each column from left to right represents a stage in the progression of the stomatal lineage (abbreviated as in *Figure 1A*). Rows from top to bottom show expression patterns of plasma membrane (PM) marker (row 1), and reporters of SPCH (row 2, beige), MUTE (row 3, orange), FAMA[LGK] (row 4, red) and CDKA1;1 (row 5). Images are of independent GCs of adaxial cotyledons at 6, 9 or 12-dpg. (**C**) Expression of each marker (rows 1 to 5) in GCs that underwent amplifying or spacing divisions. (**D**) Guard cells exhibiting pavement cell-like lobed growth with no divisions or expression of stomatal and cell cycle reporters. Cell outlines (purple) were visualized with propidium iodide or ML1p:mCherry-RCI2A. Autofluorescence of chloroplasts (blue spheres) may be visible in some images. All images are at the same magnification. Scale bar, 10 μm.

*Figure 2. Continued on next page*

*Figure 2. Continued*

The following figure supplements are available for figure 2:

**Figure supplement 1**. Expression of cell cycle and stomatal reporters in FAMA<sup>LGK</sup> plants (*FAMAp:FAMA<sup>LGK</sup>;fama*) and amiRBR (*FAMAp:amiRBR*) mutants and examples of timelapse images for SPCH and MUTE markers.

**Figure supplement 2**. Timelapse imaging of cell fate reporters in FAMA<sup>LGK</sup> lines.

**Figure supplement 3**. Guard cells in FAMA<sup>LGK</sup> plants reiterate the stomatal developmental pathway and undergo stereotypic stomatal asymmetric cell divisions that generate the diversity in phenotype.

old plants. Expression of an additional copy of tagged RBR (RBR-CFP), however, does not alter divisions in the stomatal lineage; we observed neither arrested cells nor hyperproliferating cells (*Figure 4A–B*). In FAMA<sup>LGK</sup> plants, GCs that undergo extra divisions re-express RBR as would be expected from RBR's normal expression pattern in the early stomatal lineage (*Figure 4C–G*).

Expression of early stomatal markers indicated that FAMA<sup>LGK</sup> GCs re-acquired stomatal precursor identities, but did these cells return to an even earlier stem-cell or embryonic identity? Moreover, was a change in identity tied to failure of FAMA<sup>LGK</sup> to activate its normal downstream targets? We addressed these questions by monitoring gene expression in isolated 12-dpg cotyledons of WT (Col) and FAMA<sup>LGK</sup> plants (*Figure 5A*). Analysis of genes shown in *Figures 2 and 3* to be inappropriately up-regulated in FAMA<sup>LGK</sup> verified that a qRT-PCR-based approach could accurately assess gene expression changes (*Figure 5A*, bracket indicating stomatal precursor genes). There was a dramatic increase in expression levels for the stomatal precursor genes, but variable change in expression of mature GC genes, consistent with a situation in which the overproduction of GCs through repeated re-entry is balanced by the loss of identity of older GCs (*Figure 5A*, mature GC genes). FAMA<sup>LGK</sup> was also still capable of up-regulating several, but not all, of the direct targets reported in (*Hachez et al., 2011*) (*Figure 5A*, FAMA direct targets). When expression of shoot meristem (*SHOOT MERISTEMLESS, STM*), root meristem (*WUS HOMEOBOX, WOX5*) or embryo genes (*WOX9, WOX2, FUSCA3 (FUS3), LEC1*) (*Breuninger et al., 2008*; *De Smet et al., 2010*) was monitored in FAMA<sup>LGK</sup> plants, however, we found no evidence that cells were being reprogrammed into embryonic or other stem-cell-like fates (*Figure 5A* and *Figure 5—figure supplement 1*). Taken together, the gene expression data indicate that disruption of FAMA-RBR interaction via the FAMA<sup>LGK</sup> modification leads to a stomatal lineage-specific loss of terminal commitment.

By what molecular mechanism might this specific loss of commitment take place? Analysis of chromatin states in maturing leaves revealed H3K27me3 (a chromatin mark associated with transcriptional repression) in the genomic regions of *SPCH, MUTE, FAMA, EPF1* and other stomatal genes (*Lafos et al., 2011*), and a recent report showed that manipulation of a member of the POLYCOMB REPRESSIVE COMPLEX 2 (PRC2) can alter developmental regulation of H3K27me3 deposition at *SPCH* and *MUTE* loci (*Lee et al., 2014b*). Animal and plant Rb/RBR proteins can serve as interaction bridges between chromatin modifying enzymes and specific genomic contexts (*Burkhart and Sage, 2008*; *Gutzat et al., 2012*) and RBR was previously found to be associated with SPCH regulatory regions via ChIP in whole seedlings (*Weimer et al., 2012*). Therefore it is plausible that, as the final master regulator bHLH in the stomatal pathway, FAMA (with RBR) ensures terminal differentiation of GCs by facilitating stable repression of early stomatal lineage genes. To test this model, we assayed the co-association of FAMA and RBR with regulatory regions of three key stomatal lineage genes that have significant H3K27me3 coverage (*SPCH, FAMA* and *EPF1*) and, as specificity controls, two cell cycle genes known to be RBR targets (*Nowack et al., 2012*). Because RBR is essential and expressed in nearly all cells, to accurately assay its role as a potential partner of FAMA in the stomatal lineage, we generated a Myc-tagged version of RBR expressed under the FAMA promoter (*FAMAp:RBR-MYC*). We confirmed that expression of this transgene did not alter stomatal development and that we could effectively immunoprecipitate it from plants (*Figure 5—figure supplement 2*). In ChIP assays, *SPCH, FAMA* and *EPF1* were all targets of FAMA and of stomatal lineage-expressed RBR (*Figure 5B–C* and *Figure 5—figure supplement 3*). Further dissection of the binding regions at *SPCH* and *EPF1* indicates that FAMA and RBR are enriched in the same pattern, suggesting that they bind as part of the same complex (*Figure 5—figure supplement 4*). We then tested the key prediction of our model–that association of

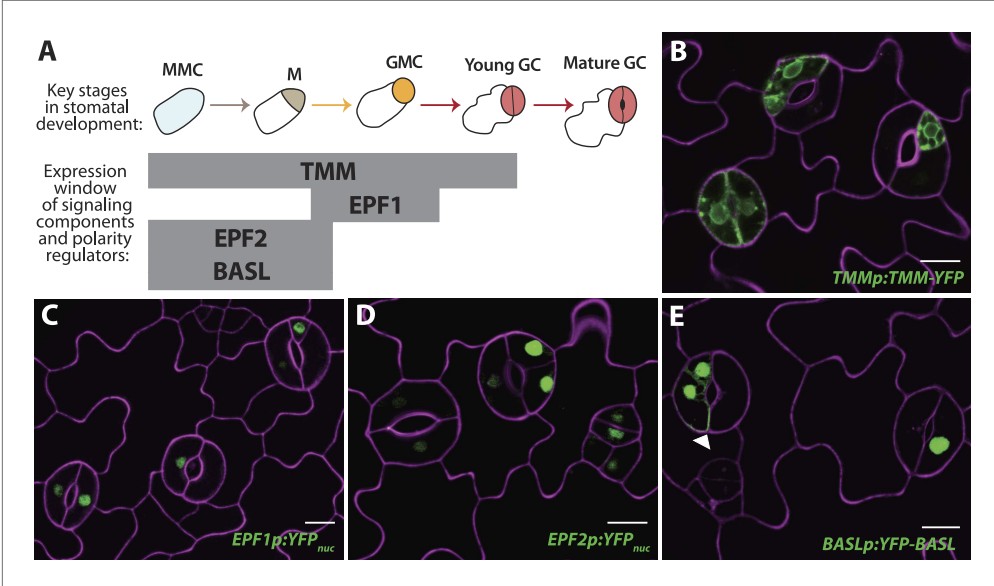

**Figure 3**. Reprogrammed FAMA^LGK guard cells re-express early stomatal signaling components and polarity regulators. (**A**) Diagram of stages of stomatal development (abbreviated as in **Figure 1A**) and expression window of signaling and polarity regulators indicated as bars spanning lineage stages. Re-expression of TMMp:TMM-YFP (**B**), EPF1p:YFP$_{nuc}$ (**C**), EPF2p:YFP$_{nuc}$ (**D**), and BASLp:YFP-BASL (**E**) in GCs from adaxial cotyledons of 6-dpg FAMA^LGK seedlings. Arrowhead in (**E**) indicates the polarized crescent characteristic of BASL in asymmetrically dividing cells. Cell outlines (purple) were visualized with propidium iodide. Scale bar, 10 μm.

RBR with a stomatal target gene is dependent on its interaction with FAMA. ChIPs of FAMAp:RBR-MYC in a FAMA^LGK background showed that RBR enrichment at the promoters of *SPCH* and *FAMA*, but not of the general RBR target gene *PCNA*, was significantly reduced (**Figure 5D** and **Figure 5—figure supplement 4**), consistent with our recruitment model. RBR enrichment at the promoter of a negative regulator of stomatal development, *EPF1*, was more variable in our assays, sometimes showing little change (**Figure 5D**), but failing to associate with RBR in other replicates (**Figure 5—figure supplement 3C**).

The association of FAMA and RBR with the *SPCH* locus is intriguing, as in normal development SPCH is required for initiation of the stomatal lineage and is essential for robust expression of all stomatal genes so far reported (**MacAlister et al., 2007**; **Pillitteri et al., 2007**; **Kanaoka et al., 2008**). In theory, failure of FAMA^LGK to stably repress *SPCH* expression could, by itself, be sufficient to reinitiate the stomatal lineage program. If this were true, ectopic expression of SPCH in GCs should recapitulate the FAMA^LGK phenotype. Expression of FAMAp:SPCH-YFP (or its hyperactive variants FAMAp:SPCH1-4A or FAMAp:SPCH2-4A [**Lampard et al., 2008**]) in an otherwise WT background, however, did not mimic FAMA^LGK (**Figure 5E**). This suggests that competence to reinitiate the stomatal pathway requires more than expression of a single 'trigger' gene, but rather a more generally permissive expression state, a fact supporting a broader chromatin regulating role for the FAMA-RBR complex (**Figure 5F**).

## Discussion

Recently, physical associations between RBR and the *Arabidopsis* transcription factor SCARECROW (SCR) were found to be essential for modulating stem-cell behavior in the root (**Cruz-Ramirez et al., 2012**, **2013**). Both SCR and FAMA bind to RBR via an LxCxE motif, yet the consequences of these transcription factor/RBR interactions are different; RBR antagonizes SCR function in asymmetric division in the root stem cell compartment (**Cruz-Ramirez et al., 2012**, **2013**), whereas RBR and FAMA have similar functions promoting differentiation at the terminal stage of stomatal development. Yet, while different RBR/transcription factor complexes may be customized for unique developmental contexts, the underlying molecular mechanisms of gene regulation might be similar. As with FAMA targets, RBR is required for repression of SCR target genes and can associate with their promoter

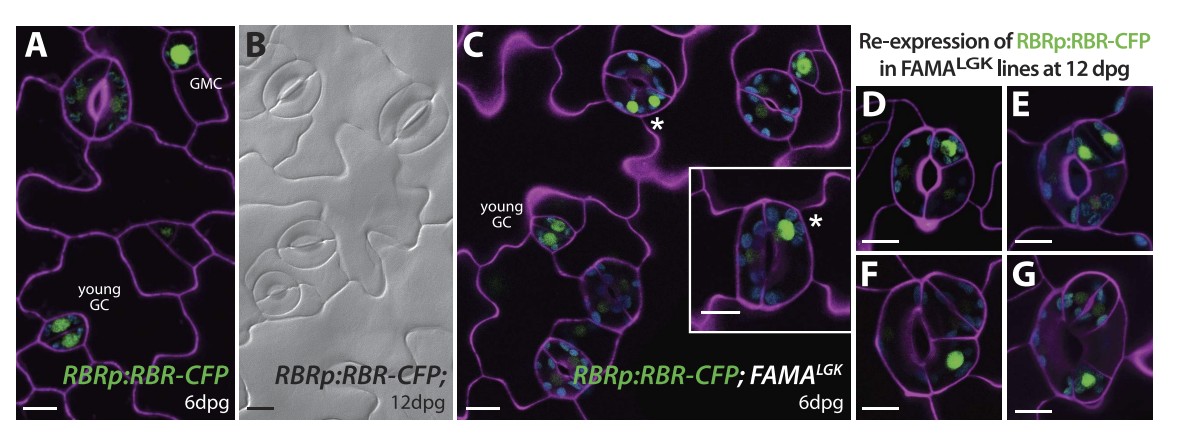

**Figure 4**. Expression of RBRp:RBR-CFP reappears in reprogramed FAMA[LGK] guard cells. RBRp:RBR-CFP (green) is expressed in GMCs and young GCs, but expression in WT does not confer any guard cell phenotype at 6-dpg (**A**) or 12-dpg (**B**). Reprogrammed guard cells in FAMA[LGK] plants re-express RBR in specific cells (green) as they recapitulate the stomatal development pathway and undergo precursor divisions. Meristemoid mother cell (MMC) and meristemoid (M) divisions (asterisks) captured at 6-dpg (**C**) and GMC and spacing asymmetric cell division (ACD) captured at 12-dpg (**D–G**). Cell outlines in confocal images are visualized with propidium iodide (purple). Small disks visible in color in these cells are chloroplasts. Scale bar, 10 μm.

regions, but there have been no experiments addressing whether disrupting association of SCR and RBR affects either proteins' chromatin association. In this study, we provide key data in support of a specific molecular mechanism for transcriptional repression utilizing RBR in combination with cell-type specific transcription factors: first, we demonstrate, through cell-type specific ChIPs, that RBR is associated with the promoter of the stomatal lineage initiator *SPCH* in cells as they are committing to terminal fates and second, we show that this binding is reduced when RBR's interaction with FAMA is disrupted. Thus, our data provide strong support for RBR being recruited by cell-type specific transcription factors to lead to transcriptional repression of their targets (*Figure 5F*).

We observed significant changes in RBR association with *SPCH* and *FAMA* regulatory regions in FAMA[LGK] lines; however, RBR was still associated with the *EPF1* locus in some experimental replicates. This could indicate that there are FAMA-independent ways to recruit RBR to this site, or that our FAMA[LGK] manipulation does not eliminate all FAMA-RBR interactions in the endogenous complex. It is interesting, however, that EPF1 differs from SPCH and FAMA in being a repressor of stomatal development and thus alleviation of the repression of *EPF1* would be expected to antagonize reprogramming to a stomatal precursor identity.

The role of PRC2 complex protein CURLY LEAF (CLF) was recently investigated in connection to stomatal lineage termination and was found to promote the accumulation of H3K27me3 marks on early stomatal lineage genes (*Lee et al., 2014b*). These data complement ours in connecting chromatin modification to stable acquisition of terminal cell identities in the stomatal lineage. Additionally, (*Lee et al., 2014b*) report phenotypes, similar to, but milder than those seen in FAMA[LGK], caused by prolonged expression of a C-terminal GFP-tagged version of FAMA (FAMA[trans]). In a timepoint and tissue common to their data and this study (12 day old cotyledons), FAMA[LGK] plants display reprogramming of ~80% guard cells (*Figure 1L*) compared with 10% of FAMA[trans]. Lee et al. interpret reinitiation of divisions in guard cells as resulting from a gain of FAMA function, however, this interpretation is at odds with previously published data that FAMA overexpression limits cell division (*Ohashi-Ito and Bergmann, 2006*; *Hachez et al., 2011*) and is difficult to reconcile with most models of chromatin-mediated repression of transcription. In light of our data showing that a loss of a specific FAMA activity (RBR binding) produces strong lineage reprogramming, we think a more parsimonious explanation of the FAMA[trans]-induced phenotype is that blocking of the FAMA C-terminus by addition of GFP creates a protein that dominantly interferes with FAMA-RBR interactions.

By independently manipulating RBR levels and RBR-FAMA interactions in terminally differentiating GCs, we could uncouple division and fate modulating roles of RBR. Notably, depletion of RBR in many contexts leads to hyperproliferation and derepression of cell cycle promoting genes (*Figure 1E* and *Figure 2—figure supplement 1* and *Borghi et al., 2010*; *Wachsman et al., 2011*; *Weimer et al., 2012*).

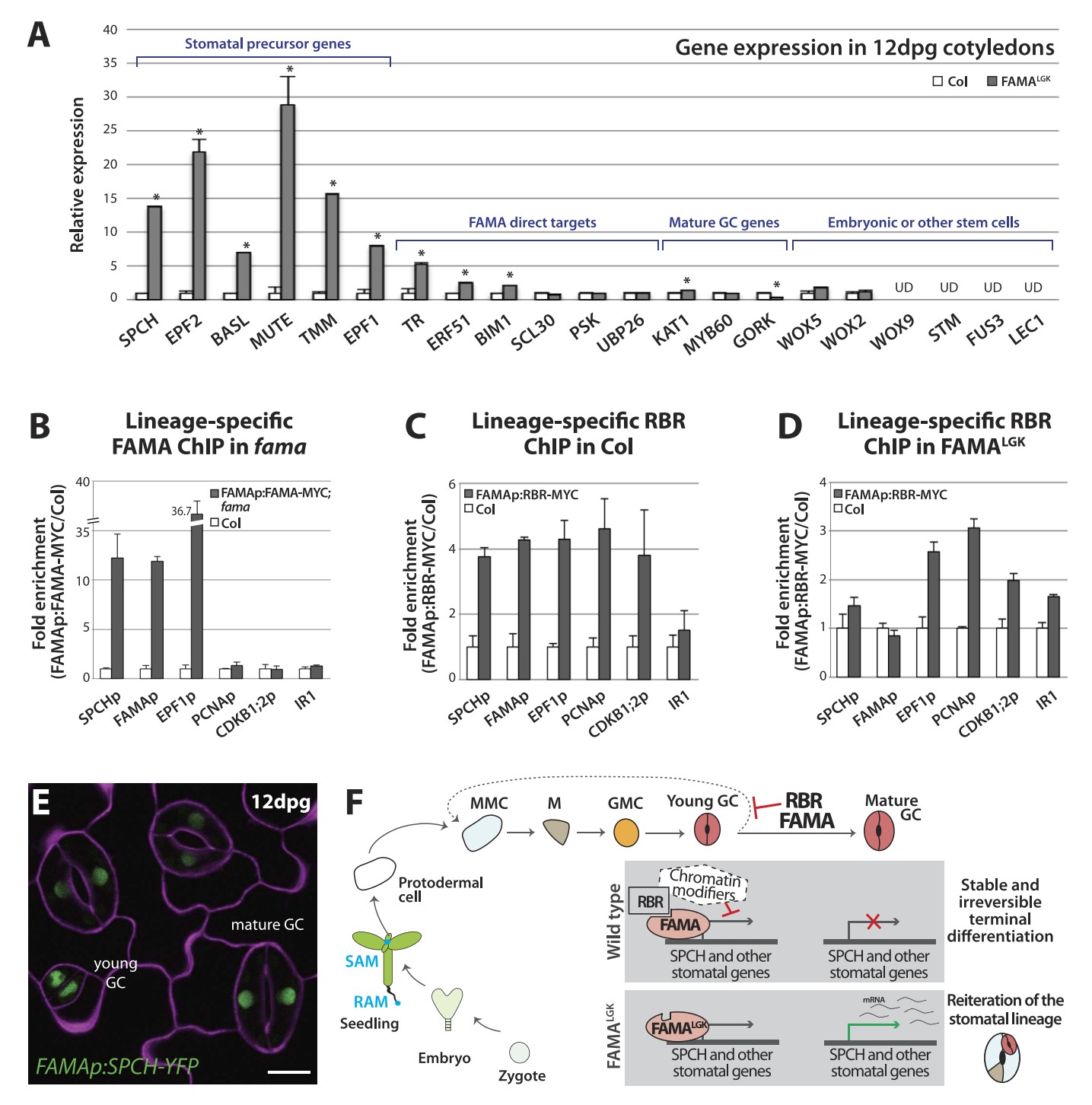

**Figure 5**. Terminal differentiation of guard cells may be mediated by FAMA-guided recruitment of RBR to suppress stomatal regulatory genes. (**A**) Expression analysis in mature cotyledons (12-dpg) of FAMA[LGK] and wild type (Col) by quantitative RT-PCR. Signals were normalized to ACTIN2 and then to Col. Values shown are means ± SEM. UD, undetected. Asterisks indicate significant difference (Student's $t$ test, * $p < 0.05$). (**B–D**) Binding of FAMA and RBR to regulatory regions of stomatal genes. ChIP assays were performed with FAMAp:FAMA-MYC in *fama* (**B**), FAMAp:RBR-MYC in Col (**C**), and FAMAp:RBR-MYC in FAMA[LGK] (*FAMAp:FAMA[LGK];fama*) (**D**) using an anti-Myc antibody as in (*Lau et al., 2014*). ChIPed DNA was quantified by qPCR with primers specific to the indicated gene promoters or the negative control region, IR1 (*Cruz-Ramirez et al., 2012*). Input-adjusted signals were normalized to Col. Values are means ± SEM. (**E**) FAMA promoter-driven expression of SPCH in wild type is not sufficient to reprogram guard cells to FAMA[LGK] phenotype. Confocal image of FAMAp:SPCH-YFP (green) in 12-dpg cotyledon visualized with propidium iodide (purple). Scale bar, 10 μm.
*Figure 5. Continued on next page*

*Figure 5. Continued*

(**F**) The stomatal lineage represents a stem-cell like lineage that is distinct from other stem-cell like compartments in the shoot, root or embryo. The FAMA-RBR module maintains terminal differentiation of guard cells (GCs) through repression of the early stomatal lineage genes, likely made permanent by chromatin modification. In FAMA$^{LGK}$ plants, RBR is no longer recruited to *SPCH* and other stomatal lineage gene promoters allowing inappropriate re-expression of these genes and subsequent reiteration of the stomatal development pathway.

The following figure supplements are available for figure 5:

**Figure supplement 1**. Validation of primers for the stem cell markers FUS3, LEC1, STM and WOX9.

**Figure supplement 2**. Generation of transgenic lines expressing Myc-tagged RBR driven by FAMA promoter, in vivo immunoprecipitation of the RBR-Myc protein and phenotypic analysis of the transgenic lines.

**Figure supplement 3**. Biological replicates for ChIP experiments in *Figure 5*.

**Figure supplement 4**. Dissection of FAMA and RBR binding on stomatal target genes.

This is in contrast to phenotypes that dominate when its association with FAMA is disrupted; namely that specific cell fates and division behaviors and their accompanying gene expression patterns re-emerge in an orderly pattern. RBRp:RBR-CFP itself becomes ectopically expressed in re-dividing FAMA$^{LGK}$ GCs. Were RBR to be playing its cell-cycle repressive role in this context, one would expect this ectopic expression might completely halt divisions. That the opposite occurs, however, suggests a qualitatively different role for RBR in combination with FAMA in these cells. Because *RBR* is expressed in all cells of the plant, it is difficult to measure whether our cell-type specific amiRNA completely eliminated *RBR* in guard cells, but our data suggest that RBR's cell-cycle repression activity is more sensitive to dosage than its activity modulating stomatal gene expression. Only by retaining RBR levels but disrupting the FAMA-dependent activity was a clear role for RBR in terminal differentiation unmasked.

In the balance between proliferation and differentiation, between developmental flexibility and robust fate commitment, several decades of cell culture and animal genetic knockout studies have placed Rb family proteins in key, if disputed, roles (*Chinnam and Goodrich, 2011*; *Gu et al., 1993*; *Sage, 2012*). Regulation of the stomatal lineage can parallel that of stem cell populations in animals at cellular, developmental and molecular levels. Both plant stomatal and mammalian myogenic lineages, for example, employ series of paralogous bHLHs during fate specification and differentiation, and activities of these bHLHs are regulated through conserved upstream kinases and by association with Rb/RBR (reviewed in *Matos and Bergmann, 2014*). Bound by immobile cell walls, stomatal lineage cells leave a record of their fate and division history in their marker expression and spatial arrangement on the leaf surface. This plant model, therefore, provides a unique opportunity to dissect cell division and cell fate activities of Rb and other conserved proteins during programming and reprogramming and is a powerful comparative system for future discoveries of fundamental regulatory mechanisms of stem cell initiation, maintenance and termination.

## Materials and methods

### Plant material and growth conditions

*Arabidopsis thaliana* Columbia-0 (Col) was used as wild type in all experiments. All mutants and transgenic lines tested are in this ecotype. The following previously described mutants and reporter lines were used in this study: *fama-1* and *FAMAp:GFP-FAMA* (*Ohashi-Ito and Bergmann, 2006*); *MUTEp:MUTE-GFP* (*Pillitteri et al., 2007*); *KAT1p:GUS* (*Nakamura et al., 1995*); *CDKB1;1p:GUS* (*Boudolf et al., 2004*); *CDC6p:GUS* (*Castellano et al., 2001*), *RBRp:RBR-CFP* (*Cruz-Ramirez et al., 2012*) and *CDKA;1p:YFP-DB* (*Jakoby et al., 2006*). The CDKA;1 reporter contains a destruction box (DB) within the YFP fusion to ensure that reporter expression does not persist after a cell division. Versions of reporters previously published with different fluorescent proteins include: *SPCHp:SPCH-YFP* (*MacAlister et al., 2007*), TMMp:TMM-YFP (*Nadeau and Sack, 2002*), EPF2p:YFP$_{nuc}$ (*Hara et al., 2009*), EPF1p:YFP$_{nuc}$ (*Hara et al., 2007*), ML1p:mCherry-RCI2A (*Roeder et al., 2010*), and *BASLp:YFP-BASL* (*Dong et al., 2009*). Seedlings were grown on 0.5 Murashige and Skoog (MS) medium at 22°C under 16 hr-light/8 hr-dark cycles and were examined at the indicated time.

## Vector construction and plant transformation

The FAMA promoter (2.5 kb, [*Ohashi-Ito and Bergmann, 2006*]) and full-length cDNAs of FAMA and RBR were cloned into Gateway compatible entry vectors, typically pENTR/D-TOPO (Life Technologies, Carlsbad, CA), to facilitate subsequent cloning into plant binary vectors. To mutate the LxCxE motif of FAMA to LxGxK, site directed mutagenesis was performed using the QuikChange II Kit (Agilent, Santa Clara, CA). Gateway entry vectors containing the FAMA promoter and FAMA$^{LGK}$ cDNA were recombined into the plant binary destination vectors pHGY (*Kubo et al., 2005*) and pGWBI (*Nakagawa et al., 2007*) to generate YFP-tagged and untagged versions of *FAMAp:FAMA$^{LGK}$*, respectively. For the *FAMAp:amiRBR* construct, the artificial microRNA sequence was designed with the Web MicroRNA Designer platform (http://wmd3.weigelworld.org). The microRNA sequence was engineered using the pRS300 plasmid as template, and together with the FAMA promoter, was subcloned into the destination vector pGWBI (*Nakagawa et al., 2007*). The constructs *FAMAp:FAMA-MYC* and *FAMAp:RBR-MYC* were generated with the tripartite recombination of the plant binary vector R4pGWB419 (*Nakagawa et al., 2008*), with the Gateway entry clones of the FAMA promoter and cDNAs of FAMA or RBR. The constructs *FAMAp:SPCH-YFP*, *FAMAp:SPCH1-4A* and *FAMAp:SPCH2-4A* were generated with the plant binary vector R4pGWB430 (*Nakagawa et al., 2008*), the FAMA promoter and the respective SPCH coding sequences described in *Lampard et al. (2008)*. Primer sequences used for each construct are provided in *Supplementary file 1*.

Transgenic plants were generated by Agrobacterium–mediated transformation (*Clough, 2005*) and transgenic seedlings were selected by growth on 0.5 MS plates supplemented with 50 mg/l hygromycin (pHGY and pGWB1 based constructs) or kanamycin 100 mg/l (pGWB419 and pGWB430 based constructs). The constructs *FAMAp:FAMA$^{LGK}$* and *FAMAp:FAMA$^{LGK}$-YFP* were transformed into the *fama/+* segregating line and homozygous lines for *fama* (as detected by PCR genotyping) were recovered in subsequent generations. *FAMAp:FAMA$^{LGK}$;fama$^{-/-}$* is referred to as FAMA$^{LGK}$ throughout the study. *FAMAp:RBR-MYC* was transformed into Col and *FAMAp:FAMA$^{LGK}$-YFP;fama* lines. All other constructs were transformed into Col.

## Quantification of phenotypic defects in FAMA$^{LGK}$

Seedlings from two independent and homozygous lines of *FAMAp:FAMA$^{LGK}$;fama* were collected at 6, 9 and 12 dpg. Samples were cleared in 7:1 ethanol:acetic acid, treated 30 min with 1 N potassium hydroxide, rinsed in water, and mounted in Hoyer's medium. Differential contrast interference (DIC) images were obtained from the middle region of adaxial epidermis of cotyledons at 20× (0.32 mm$^{-2}$ field of view) on a Leica DM2500 microscope. For quantification, 13 different guard cell phenotypes were counted and grouped into 5 classes (*Figure 1L* and *Figure 1—figure supplement 2*). Results are shown as mean percentages of each phenotypic class divided by the total number of guard cells per field view ± SEM (n = 30).

## Analysis of transcriptional and translational reporters

To analyze reporter expression in FAMA$^{LGK}$, *FAMAp:amiRBR* and *fama*, all transcriptional and translational reporters were introgressed into the mutant backgrounds and homozygous lines (as determined by PCR-based genotyping and segregation ratios) in subsequent generations were recovered for analysis. For confocal microscopy, images were taken with a Leica SP5 microscope and processed in ImageJ. Cell outlines were visualized by either 0.1 mg/ml propidium iodide in water (Molecular Probes, P3566) or the plasma membrane marker ML1p:mCherry-RCI2A. GUS staining of transcriptional reporters was performed as described in *Scarpella et al. (2004)* and seedlings were mounted in Hoyer's and visualized with DIC microscopy as described above.

## Timelapse imaging

After 6 days of growth on half strength MS media, seedlings were transferred to a sterilized perfusion chamber similar to that described in *Robinson et al. (2011)* for imaging on a Leica SP5 Confocal microscope. The chamber was perfused with ¼ strength 0.75% (wt/vol) sucrose liquid MS growth media (pH 5.8) at a rate of 2 ml/hr. Z-stacks through the epidermis of the reporter lines were captured with Leica software every 20 min (SPCH) or 2 hr (MUTE) and then processed with Fiji/ImageJ (NIH).

## Bimolecular Fluorescence Complementation (BiFC) assays

Full-length ORFs with no stop codon of each test candidate (FAMA, FAMA$^{LGK}$, bHLH93, RBR, CYCD and CYCD$^{LGK}$) were cloned into BiFC vectors (*Walter et al., 2004*) to generate fusion proteins with either N or C terminal half of the yellow fluorescence protein (YFP) fused to the C-terminus of the test

candidate. FAMA and bHLH93 constructs were reported in (*Ohashi-Ito and Bergmann, 2006*). Assays were performed in *Nicotiana benthamiana* leaves as described in *Ohashi-Ito and Bergmann (2006)*. BiFC signals were visualized on a Leica DM5000 fluorescence microscope and quantified as percentage of YFP-positive nuclei over total number of pavement cells in a field of view (centered on the injection site). Results from three experiments are presented in *Figure 1G*.

## Yeast two-hybrid assays

Full-length ORFs containing stop codons for each test candidate (FAMA, FAMA$^{LGK}$, bHLH93, BASL, RBR, CYCD, CYCD$^{LGK}$) were cloned into pENTR/D-TOPO (Life Technologies) and then recombined into the yeast vectors pGADT7 (Clontech, Mountain View, CA) and pXDGATcy86 (*Ding et al., 2007*) Yeast stain AH109 was transformed using the Yeastmaker yeast transformation system (Clontech) according to manufacture's instructions. Pairwise interactions were tested based on growth complementation on nutritional selective media.

## Quantitative RT-PCR

Cotyledons from 10 FAMA$^{LGK}$ or Col seedlings were harvested at 12 dpg and RNA was extracted using the RNeasy plant mini kit (QIAGEN, Valencia, CA) with on-column DNAse digestion. 700 ng of total RNA was used for cDNA synthesis using oligo(dT) primers and the Supercript III First-strand cDNA synthesis kit (Life Technologies). qPCR reactions were performed on a CFX96 Real-Time PCR detection system (Bio-Rad) with the Ssofast EvaGreen Supermix (Bio-Rad, Hercules, CA). Three technical replicates were performed on each of two biological replicates. Expression values were normalized to the reference gene ACTIN2 using the $\Delta^{CT}$ method and relative expression of a target was calculated from the ratio of FAMA$^{LGK}$ to Col. All data are presented as mean ± SEM. The significance of difference between the mean values was determined using two-tailed unpaired Student's *t* test. Statistical analysis was applied to normalized $\Delta^{CT}$ values. $p < 0.05$ was considered statistically significant. All calculations were performed using GraphPad Prism software. Primer sequences are listed in *Supplementary file 1*. Since expression of the embryonic genes *WOX9*, *LEC1*, *FUS3* and the shoot apical meristem gene *STM* were not detectable in cotyledons of either Col nor FAMA$^{LGK}$, we confirmed that primers were functional by testing them in RT-PCRs with RNA from immature siliques, as described in *Onate-Sanchez and Vicente-Carbajosa (2008)*. *STM*, *WOX9*, *LEC1* and *FUS3* were all detectable in these assays (*Figure 5—figure supplement 1*).

## Chromatin immunoprecipitation (ChIP) assays

ChIP experiments were carried out based on standard protocols (*Gendrel et al., 2005*) or with adaptations as described in *Lau et al. (2014)*. Briefly, for ChIPs of FAMA, ~5 g of 5-day-old whole seedlings of FAMAp:FAMA-MYC (in Col or in *fama*) and Col (control) were used as starting materials. For ChIPs of RBR, ~25 g of 5-day-old whole seedlings of Col (control) and FAMAp:RBR-MYC in Col or in FAMApro::FAMA$^{LGK}$-YFP;*fama* were used in the assays. For RBR ChIP, input materials were processed in standard-sized aliquots during nuclei isolation and DNA fragmentation steps before combining for immunoprecipitation. Expression and pull-down of the cell-type specific RBR-Myc were verified by Western and immunoprecipitation experiments (*Figure 5—figure supplement 2A–B*). Chromatin was fragmented by a Bioruptor (Diagenode) programed at high intensity for 3 × 7.5 min (cycles of 30 s on and 30 s off) at 4°C. Immunoprecipitation was carried out with a monoclonal anti-Myc antibody (71D10; Cell Signaling Technology), followed by incubation with magnetic beads (Dynabeads Protein A; Invitrogen). ChIPed DNA was purified by the ChIP DNA Clean & Concentrator (Zymo). For real-time qPCR, reactions were performed using SsoFast EvaGreen or SsoAdvanced Universal SYBR Green Supermix (Bio-Rad), according to manufacturer's recommended conditions, with primers targeted to the indicated region of selected genes (*Supplementary file 1*) on a CFX96 Real-Time PCR detection system (Bio-Rad). CT values were obtained for sonicated chromatin taken before (input) and after immunoprecipitation (ChIP). Three technical replicates were assayed for each sample. CT values for ChIP DNA were normalized to mean of CT values for input DNA (CT ChIP—µCT Input). Fold enrichment was calculated by dividing the normalized value of Myc-tagged with that of untagged Col. All data are presented as mean ± SEM. Two biological replicates were assayed for each ChIP-qPCR experiment.

## Acknowledgements

We thank members of our lab for discussions and Cuauhtémoc García-García for statistics advice. This work was funded by a National Institute of Health Grant 1R01GM086632. JLM was supported

by the Charles Yanofsky Graduate Fellowship. CH was funded by the Belgian American Educational Foundation and the Belgian National Fund for Scientific Research. OSL was funded by the Croucher Foundation. DCB is a Gordon and Betty Moore Investigator of the Howard Hughes Medical Institute.

## Additional information

### Competing interests

DCB: Reviewing editor, *eLife.* The other authors declare that no competing interests exist.

### Funding

| Funder | Grant reference number | Author |
|---|---|---|
| National Institute of General Medical Sciences | 1R01GM086632 | Juliana L Matos, On Sun Lau, Charles Hachez |
| Howard Hughes Medical Institute | | Dominique C Bergmann |
| Croucher Foundation | | On Sun Lau |
| Fonds De La Recherche Scientifique | | Charles Hachez |
| Belgian American Educational Foundation | | Charles Hachez |
| Gordon and Betty Moore Foundation | | Dominique C Bergmann |

The funders had no role in study design, data collection and interpretation, or the decision to submit the work for publication.

### Author contributions

JLM, OSL, CH, Conception and design, Acquisition of data, Analysis and interpretation of data, Drafting or revising the article; AC-R, BS, Contributed unpublished essential data or reagents, Drafting or revising the article; DCB, Conception and design, Analysis and interpretation of data, Drafting or revising the article

### Author ORCIDs

Juliana L Matos, http://orcid.org/0000-0001-9347-304X
On Sun Lau, http://orcid.org/0000-0002-8121-235X

## Additional files

### Supplementary file

• Supplementary file 1. Primers used in FAMA-RBR study.

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
