## [Decision Letter]

Thank you for sending your work entitled “Irreversible fate commitment in the Arabidopsis stomatal lineage requires a FAMA and RETINOBLASTOMA-RELATED module” for consideration at *eLife*. Your article has been favorably evaluated by Detlef Weigel (Senior editor), Richard Amasino (Reviewing editor), and 2 reviewers.

The Reviewing editor and the reviewers discussed their comments before we reached this decision, and the Reviewing editor has assembled the following comments to help you prepare a revised submission.

1) Regarding the ChIP experiments, it would be useful to do some further analyses at higher resolution. It is unclear if the target sites amplified by qPCR overlap for RBR and FAMA at the present resolution. Furthermore, an important control that should be done for at least a few of the target genes, is to generate ChIP data across several neighboring regions to ensure that the peak is real and decaying as expected.

2) In addition, please show the results from at least two biological replicates (i.e., double the number of columns in the relevant histograms).

3) Regarding the statistical treatment of the ChIP-qPCR data, rather than apply standard statistics to ratios (i.e., the normalized values), it is more appropriate to compare ratios in the ChIP-qPCR assays (tagged sample vs Col-0 control) by multiplying the standard errors of the individual results. (The errors seem small enough that the differences will hopefully still be significant, but it would be good to know that this is the case.) An ANOVA is also possible (with the CT values themselves) as is a linear regression with the Ct of the sample plotted against the Ct of the control to enable a determination of the correlation of the Ct of Col against the Ct of the control (and a p-value can be obtained for the linear regression).

4) The arguments for the statement “we largely uncoupled division and fate modulating roles of RBR” and the claim that cells do not return to an embryonic identity or earlier stem cell identity need to be more thoroughly discussed. Furthermore, the relatively easy experiment of evaluating TMO7 and WOX2 expression in marker lines to support the above statements ought to be done.

5) It is important to note whether or not pRBR-RBR-CFP rescues the rbr null mutants.

6) Regarding the claim that your data demonstrate the dosage sensitivity of RBR's cell-cycle repression activity, it is important to show that RBR levels are unaltered in FAMA-LGK mutants. Also, given the highly specific RBR accumulation pattern, qRT may not be sufficient to address this issue. Preferable, an RBR-reporter line would be used.

7) Regarding the pFAMA-RBR-Myc line that is used for their lineage-specific ChIP, it is important to note whether or not expression of this construct affects stomatal development because differentiation appears to be highly dependent on RBR dosage.

8) Regarding “These results reinforce parallels between stem-cell decisions in plants and animals at molecular and organizational levels...” There are clearly parallels in how plants and animals organize stem cells, but it is not clear what mechanism in particular you have identified in this study that reinforces this idea. This part should be further developed or removed.

---

## [Author Response]

*1) Regarding the ChIP experiments, it would be useful to do some further analyses at higher resolution. It is unclear if the target sites amplified by qPCR overlap for RBR and FAMA at the present resolution. Furthermore, an important control that should be done for at least a few of the target genes, is to generate ChIP data across several neighboring regions to ensure that the peak is real and decaying as expected*.

We have generated new ChIP data for FAMA and RBR on several regions of the *SPCH* and *EPF1* loci and show that the binding peaks are restricted to the proximal promoter regions of these genes and decay as expected. The FAMA binding peak is sharper than that of RBR, but the overlap is strong. These studies suggest that FAMA and RBR do indeed bind to the same region and are provided in Figure 5—figure supplement 3).

*2) In addition, please show the results from at least two biological replicates (i.e., double the number of columns in the relevant histograms)*.

We have repeated all of the ChIP experiments starting from new plant material. We found that in the context of this figure, showing both replicates on the same axes made them very difficult to read. Instead, we provide results from the biological replicates in Figure 5—figure supplement 4. These replicates show similar enrichments to the original ones, with the exception of RBR association with *EPF1* in a FAMA^LGK^ background (in the new replicate, RBR binding is also reduced here). We address this variation in the text.

*3) Regarding the statistical treatment of the ChIP-qPCR data, rather than apply standard statistics to ratios (i.e., the normalized values), it is more appropriate to compare ratios in the ChIP-qPCR assays (tagged sample vs Col-0 control) by multiplying the standard errors of the individual results. (The errors seem small enough that the differences will hopefully still be significant, but it would be good to know that this is the case.) An ANOVA is also possible (with the CT values themselves) as is a linear regression with the Ct of the sample plotted against the Ct of the control to enable a determination of the correlation of the Ct of Col against the Ct of the control (and a p-value can be obtained for the linear regression)*.

Thank you for this very thought-provoking comment. These calculations were new to us, so in our attempts to educate ourselves about statistical treatment of ChIP-PCR data, we consulted with colleagues and gathered information from publications. We found that statistical tests on ChIP-qPCR data are actually very uncommon (e.g. none performed in Sawa M et al., Science, 2007; Pruneda-Paz JL et al., Science, 2009; [45]; and Yu T et al., eLife, 2013). One possible reason is that due to the “sticky” nature of tagged samples, negative control regions may sometimes be called as significantly enriched (statistically) in ChIPed DNA from tagged samples when compared to WT samples, creating false-positives. Thus, most studies use enrichment level at the negative control regions as the baseline to assess binding events.

We debated how to make the clearest presentation of the data. To conform to current standards and avoid confusion, we removed the statistical tests in our ChIP-qPCR data altogether. We did calculate S.E.M. for the normalized CT values and included this in the figures. We think the additional ChIP data provided in this revision (at higher resolution and on neighboring regions, and with biological replicates) provide compelling data to support our conclusions about FAMA and RBR binding. We did include an expanded version for our ChIP-qPCR data analysis in the methods section: CT values were obtained for sonicated chromatin taken before (input) and after immunoprecipitation (ChIP). Three technical replicates were assayed for each sample. CT values for ChIP DNA were normalized to mean of CT values for input DNA (CT ChIP – μCT Input). Fold enrichment was calculated by dividing the normalized value of MYC-tagged with that of untagged Col. All data are presented as mean ± SEM.

However, should the editor and reviewer wish to have us keep the measurements of significance in the manuscript, we did calculate significance by t-test and included the t-test calculations here in the response (Figure 6) and also attempted to address the error propagation issue. Although the reviewer suggested doing an ANOVA instead of a t-test, since we are comparing only two independent samples (tagged vs. untagged) and all of our readings indicated that ANOVA is used when comparing three or more means for statistical significance, we choose to use the two-tailed unpaired Student’s t-test to determine whether normalized ChIP-qPCR values for MYC-tagged sample and untagged Col control were significantly different from one another. Should the reviewer and editor decide that these calculations are acceptable, it would be simple to substitute the graphs in Figure 6 (below) for those currently in Figure 5.Author response image 1.Student’s t-test analysis of ChIP-qPCR data presented in Figure 5. Statistical analysis was applied to normalized CT values. P<0.05 was considered statistically significant. Red arrows indicate P-values not considered statistically significant when error propagation of normalized CT values is taken into account.

The error propagation analysis is used to obtain the standard error of a new parameter (i.e. normalized values in our case) that is calculated from other parameters with standard errors (i.e. dCT). Bellow we show our ChIP-qPCR data calculation. We didn’t find an example in the literature for incorporation of error propagation for ChIP-qPCR data analysis. However, we did try to develop this for our analysis by consulting colleagues and J.R. Taylor, An Introduction to Error Analysis (University Science Books, 1982), Chapter 3, Propagation of uncertainties.

The propagated S.E.M. are shown in italics.

1) Mean of CT values from qPCR signals:

Input WT = **μCTiw** ± **εiw**

ChIPed WT = **μCTcw** ± **εcw**

Input Myc-tagged = **μCTim** ± **εim**

ChIPed Myc-tagged = **μCTcm** ± **εcm**

All our errors (ε) represent S.E.M.

2) Normalization of ChIPed qPCR values to input qPCR values:

NCTw = **μCTiw** – **μCTcw** ± ***εNCTw***

NCTm = **μCTim** – **μCTcm** ± ***εNCTm***

***εNCTw and εNCTm***
*can be propagated as:*

***εNCTw = εiw + εcw εNCTm***
*=*
***εim + εcm***

All our errors (ε) represent S.E.M.

3) Fold enrichment calculation:

NCTw/NCTw = 1 NCTm/NCTw = fold enrichment

*4) The arguments for the statement “we largely uncoupled division and fate modulating roles of RBR” and the claim that cells do not return to an embryonic identity or earlier stem cell identity need to be more thoroughly discussed. Furthermore, the relatively easy experiment of evaluating TMO7 and WOX2 expression in marker lines to support the above statements ought to be done*.

We have added text in the results and discussion sections to flesh out these statements more fully. In regards to embryonic identity, we, too, prefer using marker lines to obtain cellular resolution of expression patterns when possible. In the case of the experiments done in our manuscript, it is important to realize that these are in a genetic background of *fama-/-*; *FAMAp:FAMA-LGK* +/+; *ML1-RFP*; +/- or +/+, all of which come with their own antibiotic resistances. Because many markers show dose sensitivity, we prefer crossing in a common marker, and introgressing new markers into this background in not a fast process when considering all of the genotypes. For these reasons we also optimized a qPCR experiment (using all of the markers we had already confirmed as positive controls) that would let us more easily monitor other genes. As shown in Figure 4, genes that become expressed in FAMA^LGK^ “reprogrammed” cells, are significantly increased. Thus we feel that this assay has the sensitivity needed to detect transcripts of other genes, should they become expressed here.

In terms of genes to monitor “embryonic” or “meristem” identity, we initially chose a much larger number of genes described as such in publications. However, when looking at these genes in other manuscripts or in expression profile s of multiple tissues, we found that very few could actually be described as truly restricted to just one developmental stage. For example, the *TMO7* gene requested by the reviewer seems like a great choice from Dolf Weijer’s papers. But *TMO7=PRE3* and this gene is expressed in young leaves (see Figure 7 from the Bellini lab), so we would not be able to conclude from monitoring its expression whether cells had an early epidermal or meristem identity.Author response image 2.

WOX2 and WOX8 were reported to be expressed in the egg cell and in the zygote [Haecker, et al, Development, 2003]. *WOX8* is also expressed in cotyledons so we cannot use it, but *WOX2* does seem to be exclusively embryonic so we included this in our expanded qRT-PCR panel. We requested WOX2 reagents and recently received plasmid that we were able to transform into *fama*-/-; *FAMAp:FAMA*^*LGK*^ +/+; *ML1-RFP*; +/- and WT lines, but as it will be at least 4 months before we could confidently determine the expression pattern in stable T2 lines and we already found that *WOX2* was not upregulated in FAMA^LGK^ leaves by qRT-PCR (now in Figure 4), we did not think it was worthwhile to delay this paper for the extra months.

*5) It is important to note whether or not pRBR-RBR-CFP rescues the rbr null mutants*.

In this paper, we use pRBR-RBR-CFP only to demonstrate that the protein is in many cells, including the stomatal lineage. Perhaps the reviewer is making the important point that for functional studies like ChIP, one should be using a functional construct, but because our ChIP construct (FAMAp:RBR-MYC) is only expressed in the late stomatal lineage and RBR is an essential gene, we cannot assay rescue with it.

All RBR reagents were constructed with the same RBR coding sequence and this coding sequence was shown to rescue rbr in (Cruz-Ramírez et al., 2013, Plos biology) In addition, a C-terminal RFP-tagged version of RBR was shown previously to rescue the rbr mutant defects, suggesting that tagging RBR at its C-terminus, as in our study, does not impair RBR function (Ingouff et al., Plant Cell, 2006). We have included this information and the Cruz-Ramírez references in the revised text.

*6) Regarding the claim that your data demonstrate the dosage sensitivity of RBR's cell-cycle repression activity, it is important to show that RBR levels are unaltered in FAMA-LGK mutants. Also, given the highly specific RBR accumulation pattern, qRT may not be sufficient to address this issue. Preferable, an RBR-reporter line would be used*.

RBR is expressed in all cells in the plant. This comment makes us realize that with our choice of images to show that RBR is present in the stomatal lineage (Figure 1), we may have given the wrong impression that RBR was specific to only those cells (if we were to focus into another plane, you would see expression in other cells as shown in previously published work). We have reemphasized that RBR is in all young epidermal cells in the text.

We have included an image of RBRp:RBR-CFP in the FAMA^LGK^ background as Figure 4. We see no evidence for broadened expression pattern or extended expression in pavement cells, nor for diminished expression in GCs. In the reprogrammed GCs we would actually expect RBR to be up-regulated (re-expressed) as a consequence of these cell’s division behavior as well as because RBR is a target of SPCH (see Figure 8). This is what we observe (Figure 4) Importantly, we also show that neither RBRpro:RBR-CFP nor FAMApro:RBR-MYC is sufficient to repress (or promote) GC divisions in a WT background (Figure 4 and Figure 5—figure supplement 2, respectively).Author response image 3.

*7) Regarding the pFAMA-RBR-Myc line that is used for their lineage-specific ChIP, it is important to note whether or not expression of this construct affects stomatal development because differentiation appears to be highly dependent on RBR dosage*.

We have included an analysis of FAMAp:RBR-MYC phenotypes and expression in Figure 5—figure supplement 2. We find no evidence for the reiteration of early stomatal lineage behaviors in these GCs, nor do we find suppression of divisions in the GC precursor.

*8) Regarding “These results reinforce parallels between stem-cell decisions in plants and animals at molecular and organizational levels...” There are clearly parallels in how plants and animals organize stem cells, but it is not clear what mechanism in particular you have identified in this study that reinforces this idea. This part should be further developed or removed*.

We have changed the last two paragraphs of the paper to explain more about the RBR division effects. We make specific references to parallels between myogenic and stomatal lineage in terms of bHLHs, kinases and Rb/RBR and cite literature that discussed this more fully.